# Epstein-Barr virus ensures B cell survival by uniquely modulating apoptosis at early and late times after infection

Alexander M Price[1], Joanne Dai[1], Quentin Bazot[2], Luv Patel[3], Pavel A Nikitin[1], Reza Djavadian[4,5], Peter S Winter[6,7], Cristina A Salinas[1], Ashley Perkins Barry[1], Kris C Wood[6], Eric C Johannsen[4,5], Anthony Letai[3], Martin J Allday[2†], Micah A Luftig[1]*

[1]Department of Molecular Genetics and Microbiology, Center for Virology, Duke University School of Medicine, Durham, United States; [2]Molecular Virology, Division of Infectious Diseases, Department of Medicine, Imperial College London, London, United Kingdom; [3]Dana-Farber Cancer Institute, Harvard Medical School, Boston, United States; [4]McArdle Laboratory for Cancer Research, University of Wisconsin School of Medicine and Public Health, Madison, United States; [5]Department of Medicine, University of Wisconsin School of Medicine and Public Health, Madison, United States; [6]Department of Pharmacology and Cancer Biology, Duke University, Durham, United States; [7]Program in Genetics and Genomics, Duke University, Durham, United States

**Abstract** Latent Epstein-Barr virus (EBV) infection is causally linked to several human cancers. EBV expresses viral oncogenes that promote cell growth and inhibit the apoptotic response to uncontrolled proliferation. The EBV oncoprotein LMP1 constitutively activates NFκB and is critical for survival of EBV-immortalized B cells. However, during early infection EBV induces rapid B cell proliferation with low levels of LMP1 and little apoptosis. Therefore, we sought to define the mechanism of survival in the absence of LMP1/NFκB early after infection. We used BH3 profiling to query mitochondrial regulation of apoptosis and defined a transition from uninfected B cells (BCL-2) to early-infected (MCL-1/BCL-2) and immortalized cells (BFL-1). This dynamic change in B cell survival mechanisms is unique to virus-infected cells and relies on regulation of MCL-1 mitochondrial localization and BFL-1 transcription by the viral EBNA3A protein. This study defines a new role for EBNA3A in the suppression of apoptosis with implications for EBV lymphomagenesis.

*For correspondence: micah. luftig@duke.edu

†Deceased

## Introduction

An estimated 15% of cancers worldwide are caused by infectious agents like viruses. Of these, Epstein-Barr virus (EBV) was the first human tumor virus discovered and is associated with malignancies of both B lymphocyte and epithelial cell origin. Normally, T cells prevent the development of EBV-associated malignancies and the virus persists quiescently in latently-infected memory B cells. As a result, nearly all adults in the world are latently infected (*Longnecker et al., 2013*), making EBV a particularly widespread and successful pathogen. However, in the absence of a healthy immune system, such as following organ transplant or HIV infection, EBV can transform B cells leading to markedly increased rates of lymphoma in these populations.

Primary human B cell infection in vitro by EBV leads to their growth transformation and serves as a model for oncogenesis in EBV-associated lymphomas. This process requires the expression of six

**eLife digest** Over 90% of adults around the world are infected with the Epstein-Barr virus. Like other closely related viruses, such as those that cause chicken pox and cold sores, an infection lasts for the rest of the person's life, although the virus generally remains in a latent or dormant state. However, under certain conditions the latent viruses can cause cancers to develop; in fact, it is estimated that such infections are responsible for nearly 2% of all cancer deaths worldwide.

One way that healthy human cells prevent cancer is by triggering their own death in a process called apoptosis. The Epstein-Barr virus can block apoptosis, therefore making the cells more likely to become cancerous. Previous research identified one protein in the Epstein-Barr virus that promotes cancer by preventing infected cells from dying as normal. However, even in the absence of this protein, Epstein-Barr virus-infected cells remain resistant to apoptosis. This suggests that the virus has another way of blocking cell death.

Price et al. have now used a technique that stresses living cells in a way that reveals which proteins prevent apoptosis to study human cells infected with the Epstein-Barr virus. This revealed that soon after infection, the virus could force the human cell to produce MCL-1, a protein that prevents cell death. Later, the Epstein-Barr virus enlisted a second human protein called BFL-1, which makes the infected cell further resistant to apoptosis.

Price et al. discovered that a protein in the Epstein-Barr virus called EBNA3A controls the production of the MCL-1 and BFL-1 proteins. In the future, developing therapies that target these proteins may lead to new treatments for cancers caused by the Epstein-Barr virus. Such treatments would be likely to have fewer side effects for patients than traditional chemotherapies.

viral latent genes: Epstein-Barr Nuclear Antigen 1 (EBNA1), EBNA2, EBNA-LP, EBNA3A, EBNA3C and Latent Membrane Protein 1 (LMP1) (*Longnecker et al., 2013*). EBNA2, the major viral transcriptional activator and the first to be expressed upon infection, upregulates expression of the other viral EBNA genes as well as host genes (*Wang et al., 2000*; *Zimber-Strobl and Strobl, 2001*). EBNA2 also activates the expression of the viral oncoprotein LMP1, which drives the immortalization of the infected B cell into a lymphoblastoid cell line, or LCL, that proliferates indefinitely in culture. Together, the EBV latent oncoproteins mimic key aspects of normal B cell biology, promote proliferation, and inhibit the host innate tumor suppressor response to uncontrolled proliferation (*Allday, 2013*; *Price and Luftig, 2014*).

It is well-established that the LMP1 oncoprotein is necessary for continued LCL growth and survival (*Kaye et al., 1993*; *Zimber-Strobl et al., 1996*). LMP1 mimics a pro-survival TNF receptor that signals constitutively through the NFκB pathway to promote proliferation and suppress apoptosis in transformed cells (*Cahir-McFarland et al., 2000*; *Okan et al., 1995*; *Pratt et al., 2012*). In fact, inhibition of the NFκB pathway activated downstream of LMP1 leads to apoptosis in LCLs. However, recent work from our laboratory has demonstrated that, early after infection, B cells rapidly proliferate despite low levels of LMP1 expression and NFκB activation (*Nikitin et al., 2010*; *Price et al., 2012*). Surprisingly, these early-infected cells also show no overt signs of apoptosis despite strong activation of the cellular DNA damage response (*Nikitin et al., 2014*). This suggests that in early-infected cells, there is a viral-mediated mechanism independent of LMP1 signaling that promotes survival.

Several LMP1-independent mechanisms to subvert apoptosis have been described. For example, the viral BCL-2 homologue BHRF1 confers chemoresistance in a mouse model of Burkitt lymphoma (BL) (*Kvansakul et al., 2010*). Similarly, the viral BHRF1 and BALF1 proteins exert non-redundant roles in protecting initially infected cells from apoptosis (*Altmann and Hammerschmidt, 2005*). In addition, expression of the EBNA3 proteins induces resistance to apoptosis in BL cells and primary B cells by downregulating Bim expression (*Paschos et al., 2009*; *Skalska et al., 2013*), thereby making infected cells less 'primed' for apoptosis. The EBNA3 family is also known to epigenetically downregulate additional human genes involved in cell cycle regulation and apoptosis (*Allday, 2013*; *Harth-Hertle et al., 2013*; *Hertle et al., 2009*; *Paschos et al., 2012*; *Skalska et al., 2013*). It has been shown that the EBNA3 proteins mediate these epigenetic effects, in part, via interacting with

additional proteins in such a way that chromatin architecture can be dramatically altered (*Bazot et al., 2015*; *McClellan et al., 2013*). While thought to be generally repressive in nature, recent work has shown that in some instances EBNA3 proteins can activate gene transcription (*Bazot et al., 2015*). In sum, viral proteins confer several survival advantages to infected cells, which often results in EBV-positive lymphomas that are more difficult to treat than their EBV-negative counterparts (*Kelly et al., 2006*).

The intrinsic apoptosis pathway is controlled by complex protein—protein interactions at the surface of mitochondria that regulate the release of cytochrome c (*Youle and Strasser, 2008*). These functional interactions can be interrogated with an established approach called 'BH3 profiling', which uses peptides derived from BH3-only pro-apoptotic proteins to identify the anti-apoptotic BCL-2 family members important for preventing apoptosis in a particular cell (*Del Gaizo Moore and Letai, 2013*; *Deng et al., 2007*). Furthermore, molecular characterization of the interface between BH3 domains of these proteins has allowed for the creation of a new class of small molecule inhibitors that promote apoptosis directly in sensitive cell types (*Billard, 2013*; *Oltersdorf et al., 2005*). These so-called 'BH3 mimetics' are highly specific and show great promise in clinical trials (*Letai, 2008*). In this study, we characterized the BH3 profile of uninfected and EBV-infected primary human B cells from early proliferation through long-term outgrowth. We validated these profiles using BH3 mimetics and, importantly, identified a novel virus-specific transition in survival mechanisms between early and late stages of latent infection with implications for therapy of EBV-associated malignancies.

## Results

### BH3 profiling reveals two distinct stages of mitochondrial priming after EBV infection

Previously, our laboratory has identified an early period after EBV infection of primary human B cells where, despite robust virus-mediated proliferation and activation of the pro-apoptotic DNA damage response, there is little expression of the key viral survival factor LMP1 (*Price et al., 2012*). In order to define the mechanism of survival in the absence of LMP1 and NFκB activity early after infection, as well as through long-term outgrowth, we performed BH3 profiling to query apoptotic priming at the mitochondria of infected cells. Briefly, this technique works by permeabilizing cells, staining with a mitochondrial potential-sensitive dye, and then treating cells with peptides derived from the BH3 domains of pro-apoptotic BH3-only proteins, including the activator BH3-only protein BIM and the sensitizer BH3-only proteins BAD, PUMA, NOXA, BMF, and HRK (*Figure 1A*). By measuring mitochondrial depolarization induced by the activator BH3 peptide, BIM, and the pan-apoptotic sensitizer, PUMA, BH3 profiling can measure the overall 'priming' of the cells for apoptosis (that is, the cell's distance from the apoptotic threshold). Further, by measuring depolarization induced by the sensitizer BH3 peptides BAD, NOXA, BMF, and HRK, BH3 profiling can also determine the cell's specific dependence on particular anti-apoptotic BCL-2 family proteins, including BCL-2, BCL-XL, MCL-1, and BFL-1 (*Del Gaizo Moore and Letai, 2013*). We BH3 profiled three cell states: (i) uninfected B cells purified from the peripheral blood of normal human donors (B Cell), (ii) FACS-purified proliferating EBV-infected B cells seven days post-infection (Prolif), and (iii) the resulting EBV-transformed lymphoblastoid cell lines (LCL) derived five weeks post-infection from the same donors (*Figure 1B, C*). Both latently infected cell populations are 100% EBV-positive as shown previously by immunofluorescence and fluorescent in situ hybridization (*Nikitin et al., 2010*).

BH3 profiling revealed that EBV infection of primary B cells only modestly reduces overall mitochondrial priming (*Figure 1C*, *Figure 1—figure supplement 1* compare 10 μM Puma from B to Prolif and 1 μM Bim from B/Prolif to LCL). However, the BH3 profiles revealed marked changes in the BCL-2 family dependencies between the groups suggesting specific, testable hypotheses (*Figure 1D*). For instance, a cell dependent on BCL-2 or BCL-w for survival would exhibit mitochondrial depolarization upon exposure to Bad, Puma, or Bmf peptides, while a cell dependent upon MCL-1 would depolarize upon exposure to Noxa, Puma, or Bmf peptides. Uninfected B cells depolarized their mitochondria upon treatment with Bad, Puma, or Bmf peptides, but not Noxa or Hrk peptides, indicating reliance on BCL-2 or BCL-w as a pro-survival BCL2 family member (*Figure 1E*). EBV-infected proliferating B cells (Prolif) lost sensitivity to the Bad peptide but remained sensitive to

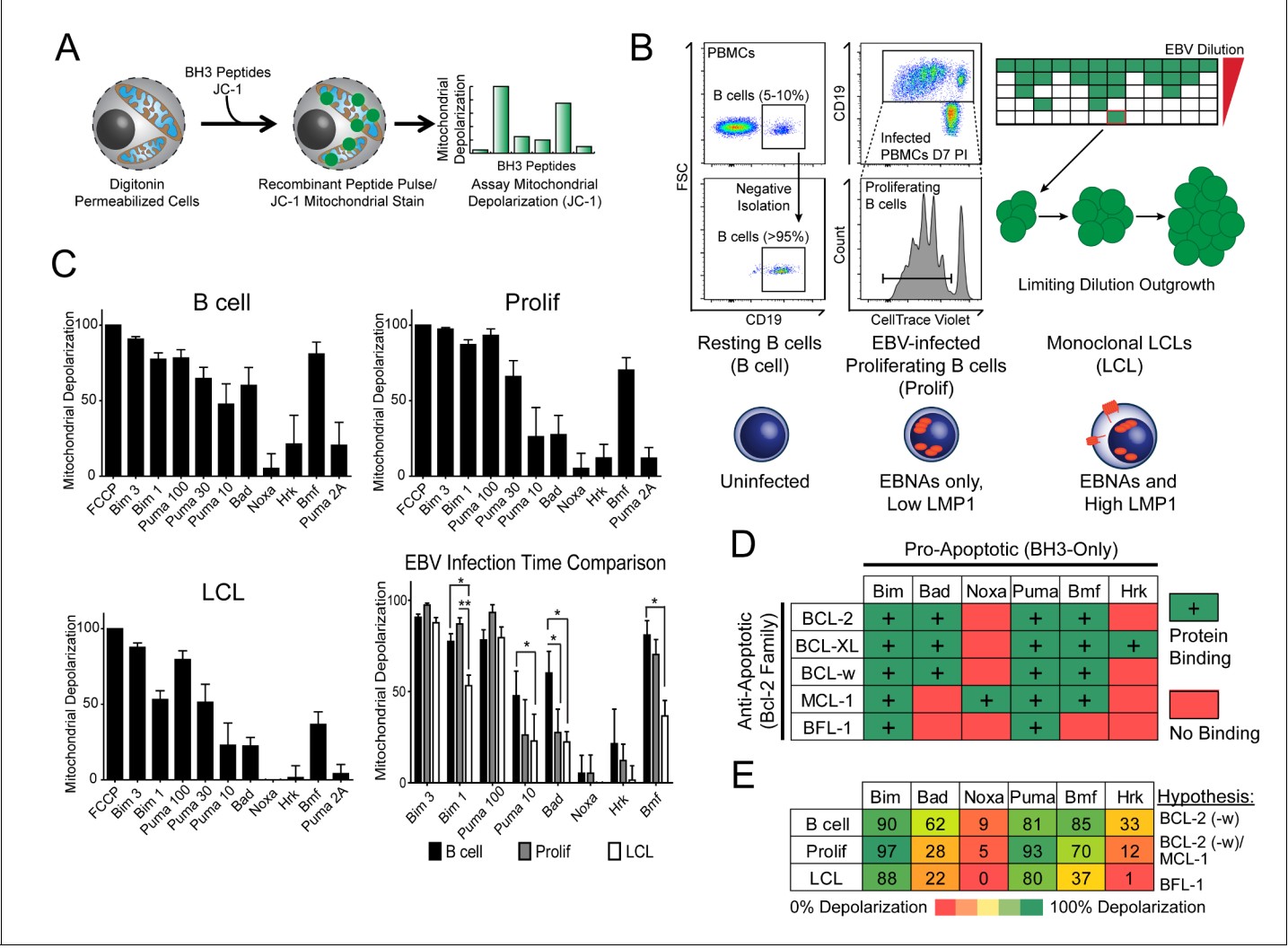

**Figure 1.** BH3 profiling reveals two distinct stages of mitochondrial priming after EBV infection. (A) Schematic of the BH3 profiling technique, which involves first permeabilizing the outer membrane followed by incubation with BH3-only peptides to induce depolarization of the mitochondrial membrane. Depolarization is measured by fluorescent JC-1 dye and quantified. (B) Schematic of EBV-mediated outgrowth of infected CD19+ B cells into a lymphoblastoid cell line (LCL). Negative isolation of peripheral blood mononuclear cells (PBMCs) yields CD19+ B cells of >95% purity. Proliferating cells (Prolif) are analyzed by flow cytometry based on the dilution of the fluorescent proliferation tracking dye CellTrace Violet and cells that have divided more than once are sorted to purity. Monoclonal LCLs are grown out from PBMCs that have been infected with a limiting dilution of EBV. (C) Analysis of the mitochondrial depolarizations from uninfected B cells, hyper-proliferating infected cells (Prolif), and LCLs when treated with 100 μM of the indicated BH3-only peptides (horizontal axis). Mitochondrial depolarization, normalized to an FCCP control, is reported as the mean value from five different donors. Error bars indicate standard error of the mean (SEM) from five matched human donors. (bottom, right) Compiled, side-by-side comparisons of mitochondrial depolarization of the three cell types and more detailed statistical results from paired t-tests are shown below. *p<0.05, **p<0.01 (D) Schematic of the selective interactions between pro-apoptotic BH3-only and anti-apoptotic BCL2 members. Green boxes and plus signs indicate protein-binding interactions that lead to mitochondrial depolarization, red boxes indicate no interactions. (E) Schematic of BH3 profiles compiled from uninfected B cells, hyper-proliferating infected cells (Prolif), and LCLs. Shown is the 10 μM Bim treatment, all other treatments are 100 μM. Numbers and color scale correspond to percentage of mitochondrial depolarization. Formulated hypotheses regarding pro-survival BCL2 members that account for each profile are in the column to the right.

The following source data and figure supplement are available for figure 1:

**Source data 1.** Source data for individual responses to BH3 peptides of uninfected, early-infected, and late-infected B cells.

**Source data 2.** Source data for individual responses to Bim 1, Puma 10, Bad, and Bmf BH3 peptides of uninfected, early-infected, and late-infected B cells.

**Figure supplement 1.** Varying sensitivity to select BH3-only peptides reveals differences in apoptotic regulation during early- and late-infection with EBV.

Puma and Bmf, supporting a survival mechanism reliant on both MCL-1, to block Bad-induced depolarization, and BCL-2/BCL-w, to block Noxa-induced depolarization (*Figure 1E*). Finally, LCLs lost sensitivity to Bmf, but sensitivity to Puma did not change between Prolif and LCLs, suggesting additional BFL-1 mediated pro-survival effects in LCLs (*Figure 1E*).

## EBV infection promotes potent resistance to BCL-2 antagonists

We hypothesized that EBV infection reduces the dependence of B cells on BCL-2, BCL-xL, or BCL-w for survival given the loss of Bad sensitivity between uninfected and early-infected B cells (*Figure 1E*). To test this, we treated PBMCs with small molecule BH3 mimetics either concurrent with EBV infection or at the onset of proliferation (3.5 days post-infection (dpi)) and measured the number of proliferating EBV-infected B cells at 7 dpi using flow cytometry and the proliferation tracking dye CellTrace Violet (*Figure 2A*). We also assayed the sensitivity of monoclonal LCLs to BH3 mimetic treatment for an equivalent period of time (*Figure 2A*).

We first tested the sensitivity of EBV-infected cells to ABT-737, a potent small molecule inhibitor of BCL-2, BCL-xL, and BCL-w (*Oltersdorf et al., 2005*). Consistent with dependence of uninfected B cells on these BCL-2 family members predicted by BH3 profiling, treatment with ABT-737 concurrent

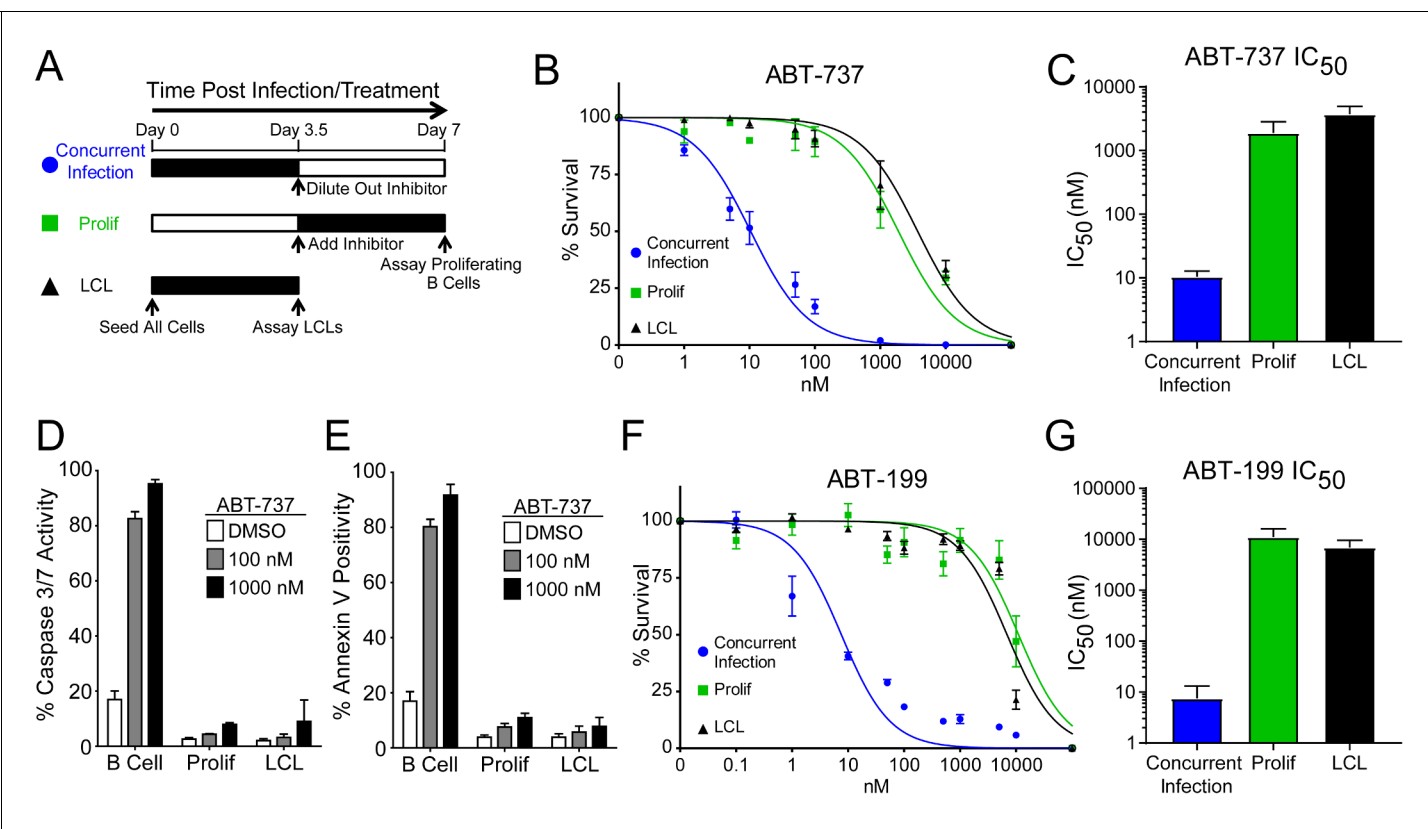

**Figure 2.** EBV Infection promotes potent resistance to BCL-2 antagonists. (A) Schematic of drug treatment time course. (B) Dose-response curves generated from treating 3–5 human donors with the BCL-2, -xL, and –w inhibitor ABT-737. Percent survival is the percent of proliferating CD19+ B cells compared to DMSO-treated controls at each time point. (C) Average $IC_{50}$ with 95% confidence intervals are plotted for ABT-737 treatment at three different times post infection. (D) Analysis of caspase activity induced by ABT-737. A fluorescent caspase 3/caspase 7 reporter was used and analyzed by FACS; values are reported as average plus SEM of three human donors. (E) Analysis of phosphatidylserine exposure induced by ABT-737. Fluorescent Annexin-V was used and analyzed by FACS; values are reported as average plus SEM of three human donors. (F) Same as (B), except the dose-response is to the BCL-2-specific inhibitor ABT-199. (G) Average $IC_{50}$ with 95% confidence intervals are plotted of ABT-199 treatment of 2–3 human donors at three different times post infection.

The following source data is available for figure 2:

**Source data 1.** Source data for cell counts and apoptosis assays performed with BH3 mimetics targeting BCL-2.

with EBV infection resulted in an IC$_{50}$ in proliferating B cells of ~10 nM (*Figure 2B–C*). However, when ABT-737 was added to EBV-infected cells after the onset of virus-induced proliferation or to immortalized LCLs we observed high-level ABT-737 resistance with an IC$_{50}$of ~2–4 µM (*Figure 2B–C*). ABT-737 promoted apoptosis as measured by caspase 3/7 activity and Annexin V positivity in uninfected B cells, while proliferating cells at 7 dpi and EBV transformed LCLs were resistant to apoptosis induction (*Figure 2D–E*).

We next sought to confirm BCL-2 as the specific anti-apoptotic protein important for uninfected B cell survival and to further validate our BH3 profiling results. To do this, we used ABT-199, a specific inhibitor of BCL-2 with minimal off-target effects on BCL-xL or BCL-w (*Souers et al., 2013*). Similar to ABT-737, EBV induced potent resistance to ABT-199 (*Figure 2F–G*). Therefore, uninfected human peripheral blood CD19+ B cells depend on BCL-2 for survival and EBV upregulates additional survival proteins to potently suppress apoptosis induced by BCL-2 antagonism.

## MCL-1 and BCL-2 protect EBV-infected proliferating B cells from apoptosis early after infection, while BFL-1 additionally protects LCLs from apoptosis late in infection

Our BH3 profiling experiments predicted that EBV-infected B cells are less primed for apoptosis due to the combined protective effects of MCL-1 and BCL-2 during early proliferation with additional protection afforded by BFL-1 in immortalized B cells (*Figure 1E*). To functionally characterize EBV-regulated survival, we first analyzed BCL-2 family member mRNA expression through B cell outgrowth. We observed that MCL-1 mRNA increased from uninfected B to EBV-infected and proliferating B cells and was further increased in LCLs (*Figure 3A*). Over the same infection time course, both BCL-2 and BFL-1 were initially downregulated in early proliferating B cells relative to uninfected B cells, then upregulated during LCL outgrowth. These data are consistent with prior transcriptomic analyses from our group (*Price et al., 2012*) and others indicating that BCL-2 and BFL-1 are both ultimately upregulated by EBV (*D'Souza et al., 2004*; *Henderson et al., 1991*). BCL-2 and MCL-1 protein levels mirror their respective mRNA expression levels during B cell outgrowth (*Figure 3B*, quantified in *Figure 3C*; note there is no viable BFL-1 antibody). MCL-1 has been known to be expressed as different isoforms, a long (MCL-1(L)) and a short (MCL-1(S)) (*Perciavalle et al., 2012*). While the MCL-1(L) isoform is associated with anti-apoptotic activity, MCL1(S) appears to be more important for mitochondrial maintenance; thus, for the scope of this study, we focused primarily on MCL-1(L), referred to as MCL-1. The aforementioned data indicate that ABT-737 resistance coincides with MCL-1 induction. We therefore sought to test the importance of MCL-1 in promoting EBV-induced ABT-737 resistance.

We hypothesized that selective inhibition of MCL-1 should render early proliferating cells, but not LCLs, sensitive to drugs that target BCL-2, namely ABT-737 and the BCL-2 specific inhibitor ABT-199. Treatment of EBV-infected proliferating B cells at 7dpi or LCLs with either ABT-737, ABT-199, or a specific MCL-1 inhibitor A1210477 (A-1210) induced minimal apoptosis (*Figure 3D–E*). However, when cells were treated with both A-1210 and either ABT-737 or ABT-199, early proliferating cells, but not LCLs, displayed elevated markers of apoptosis (*Figure 3D–E*). The combined action of ABT-199 and A-1210 supports the hypothesis generated by BH3 profiling that early-infected cells are more dependent upon MCL-1 and BCL-2 for survival than are LCLs.

Further corroborating these findings, we found that indirect inhibition of MCL-1 with flavopiridol led to sensitivity of early-infected cells to BCL-2 antagonists. MCL-1 is a highly labile transcript and protein, and its expression is known to be sensitive to RNA polymerase II stalling induced by CDK9 inhibition (*Gojo et al., 2002*). Indeed, the CDK9 inhibitor, flavopiridol, induced rapid decreases in MCL-1 mRNA and protein, while leaving BCL-2 and BFL-1 levels largely unaffected (*Figure 3—figure supplement 1A–B*). Supporting our A-1210 data, we found that early proliferating cells were more sensitive to apoptosis induced by a combination of flavopiridol and ABT-737 than LCLs (*Figure 3—figure supplement 1C*) (*Leverson et al., 2015*).

To determine if BFL-1 was important for survival in LCLs, we utilized the CRISPR/Cas9 system to induce mutations in the BFL1 (*Bcl2a1*) coding sequence. We successfully generated a clonal LCL with deletions in both BFL1 alleles as assessed by Sanger sequencing (*Figure 3F*). In one allele there is a 24 base pair (bp) in-frame deletion removing a critical loop prior to the BH3 binding cleft and the other a 29 bp deletion in the *BFL1* gene resulting in a frameshift mutation and a subsequent premature stop codon at amino acid 50 of the BFL-1 protein. These deletions were clearly evident by RT-

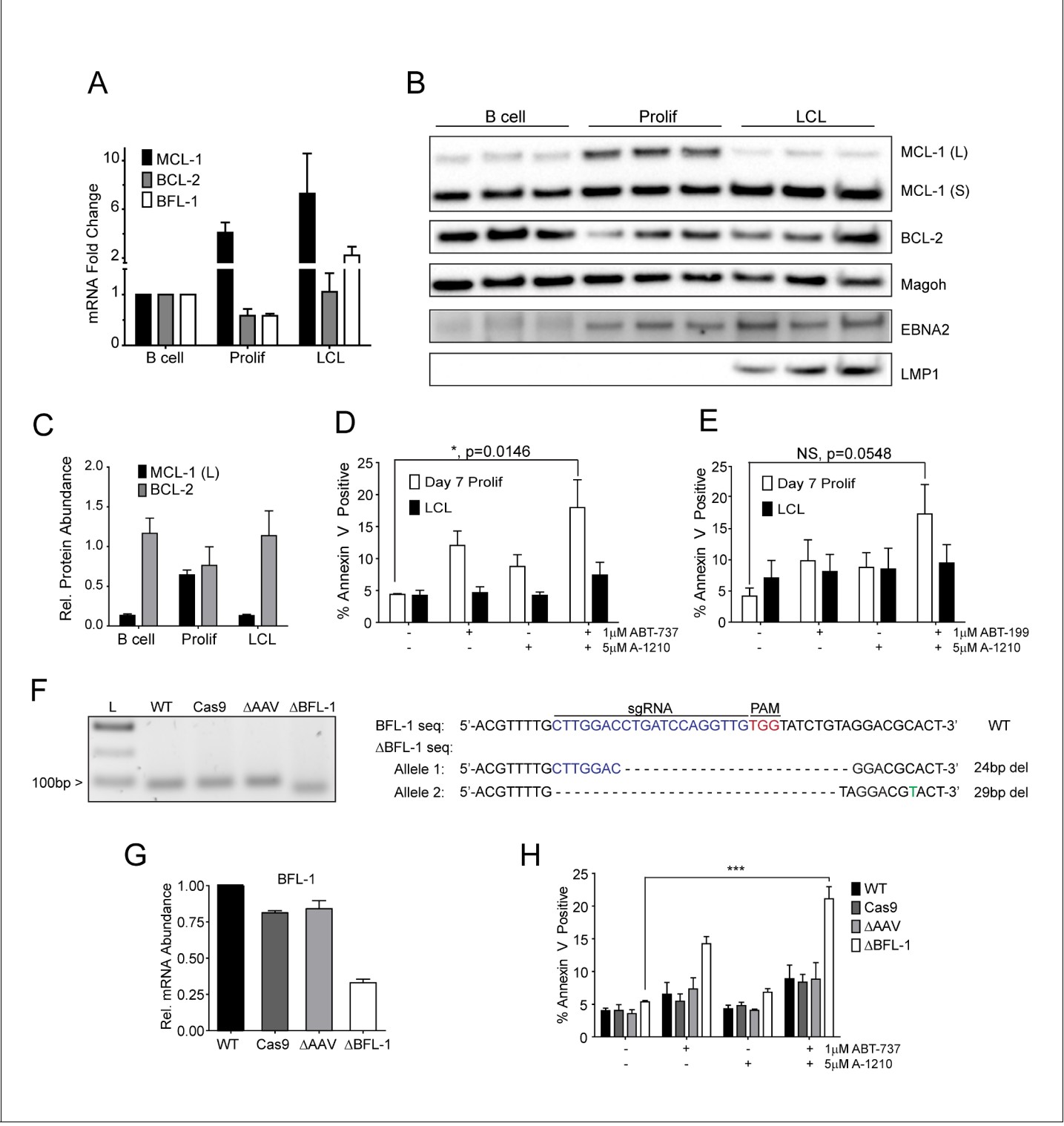

**Figure 3.** MCL-1 collaborates with BCL-2 to protect EBV-infected proliferating B cells early after infection while BFL-1 additionally protects LCLs late after infection. (**A**) Quantitative PCR (qPCR) of MCL-1, BCL-2, and BFL-1 mRNA levels post EBV infection. Average plus SEM of three human donors is plotted. (**B**) Immunoblot analysis of MCL-1 isoforms, BCL-2, EBNA2, LMP1, and Magoh (loading control) during B cell immortalization. Protein lysates from three matched donors were obtained from uninfected B cells and infected B cells sorted Day seven post-infection for proliferating cells (Prolif). Three unmatched LCLs were also included. (**C**) Protein levels of MCL-1(L) and BCL-2 from (**B**) were quantified by densitometry and normalized to the Magoh loading control. Average relative protein abundance is reported plus SEM of three donors. (**D**) Analysis of apoptosis by Annexin-V positivity in Day seven proliferating EBV-infected cells and LCLs that were treated with A1210477 (A-1210) with or without concurrent ABT-737 for 24 hr.

*Figure 3 continued on next page*

*Figure 3 continued*

Measurements were taken as a percentage of the proliferating B cell population and plotted as average plus SEM of three human donors. Data were analyzed by 2-way ANOVA, which showed a significant interaction between cell type and drug treatment (p=0.0352). *p=0.0146, by two-tailed Student's t-test. (E) Same as (D) but with ABT-199 for 24 hr. Measurements were taken as a percentage of the proliferating B cell population and plotted as average plus SEM of three human donors. Data were analyzed by 2-way ANOVA. No significant interaction between cell type and drug treatment (p=0.6288). (F) (Left) Agarose gel of RT-PCR products from ΔBFL-1 mutant cell line generated with the CRISPR/Cas9 system. Controls included are from a matched LCL wildtype (WT), Cas9-only expressing (Cas9) LCL, and Cas9-expressing LCL that was transduced with an sgRNA specific to the adeno-associated virus integration site (*AAVS1*) on the human genome (ΔAAV). Ladder (L) on the left is included. (Right) The sequences for the wildtype and mutant BFL1 sequences with the BFL1-specific sgRNA (in blue), PAM site (in red), and allele-specific SNP (in green). (G) qPCR of BFL-1 mRNA levels in WT, Cas9, ΔAAV, and ΔBFL-1 cell lines. Average plus SEM of three independent experiments are plotted. (H) Analysis of apoptosis by Annexin-V positivity in WT, Cas9, ΔAAV, and ΔBFL-1 cell lines that were treated with A-1210 with or without concurrent ABT-737 treatment. Measurements were taken as a percentage of the proliferating B cell population and plotted as average plus SEM of three independent experiments. Data were analyzed by 2-way ANOVA, which showed a significant interaction between cell type and drug treatment (p=0.0004). ***p=0.0002, by two-tailed Student's t-test.

The following source data and figure supplement are available for figure 3:

**Source data 1.** Source data for protein and mRNA expression levels of anti-apoptotic members, apoptotic assays, and characterization of a BFL1-CRISPR mutant.

**Source data 2.** Source data for *Figure 3—figure supplement 1*.

**Figure supplement 1.** Flavopiridol sensitizes EBV-infected early-proliferating B cells to ABT-737.

PCR (*Figure 3F*), and, consequently, the mutant ΔBFL-1 LCL expressed significantly reduced levels of BFL-1 mRNA as compared to WT LCL, LCL expressing Cas9 alone or Cas9-expressing LCLs targeting *AAVS1* as a negative control (*Figure 3G*). ΔBFL-1 LCLs were significantly more sensitive to treatment with a combination of ABT-737 and A-1210 relative to WT, Cas9, or *AAVS1* sgRNA control LCLs (*Figure 3H*). These data support the hypothesis defined by our BH3 profiling data that LCLs depend on BFL-1, MCL-1, and BCL-2 to protect from apoptosis induced by viral oncoprotein-driven proliferation.

## Resistance to BCL-2 antagonism is virus specific

A hallmark of B cell biology is rapid proliferation in response to antigen and cytokines leading to maturation via germinal center reactions into the memory and plasma cell lineages (*Goodnow et al., 2010*). In cell culture, mitogens such as the TLR9 ligand CpG DNA as well as T cell derived CD40 ligand and IL-4 (CD40L/IL-4) promote B cell proliferation similar to EBV infection (*Elgueta et al., 2009*; *Krieg et al., 1995*; *Nikitin et al., 2014*) (*Figure 4A–C*). To assess whether EBV-mediated ABT-737 resistance was linked to B cell proliferation per se or was specific to EBV infection, we stimulated primary B cells with CpG or CD40L/IL-4 and queried survival. We found that, while EBV induced marked ABT-737 resistance (IC$_{50}$ ~3–4 μM), both CpG and CD40L/IL-4 stimulated B cells were significantly more sensitive to ABT-737 (IC$_{50}$ ~200 nM) (*Figure 4D–E*). Consistently, mitogen-stimulated proliferating B cells had increased caspase 3/7 activity and Annexin V positivity following ABT-737 treatment (*Figure 4F–G*) while EBV-infected cells displayed only marginally increased activity above basal levels (*Figure 2D–E*). These data strongly support the hypothesis that resistance to BCL-2 antagonism is specific to EBV-induced proliferation. We next sought to characterize the EBV factors necessary for ABT-737 resistance.

## EBV-induced resistance to BCL-2 antagonism is mediated by EBNA3A

EBV requires expression of protein products from the heavily spliced Epstein-Barr Nuclear Antigen (EBNA) transcriptional unit to promote B cell proliferation and survival (*Price and Luftig, 2014*). Of key importance is EBNA2, a protein encoded initially by transcripts from the B cell-specific viral W promoter (Wp). EBNA2 subsequently activates the viral C promoter (Cp) just upstream of Wp which drives increased expression of EBNA2 as well as the other EBNAs (*Price and Luftig, 2014*; *Woisetschlaeger et al., 1991*) (*Figure 5A*). To assess the role of EBNA2 in ABT-737 resistance of early-infected cells, we used an EBNA2-deleted strain of EBV called P3HR1 (*Hinuma et al., 1967*). Since the EBNA2 deletion in P3HR1 renders the virus incapable of inducing proliferation and

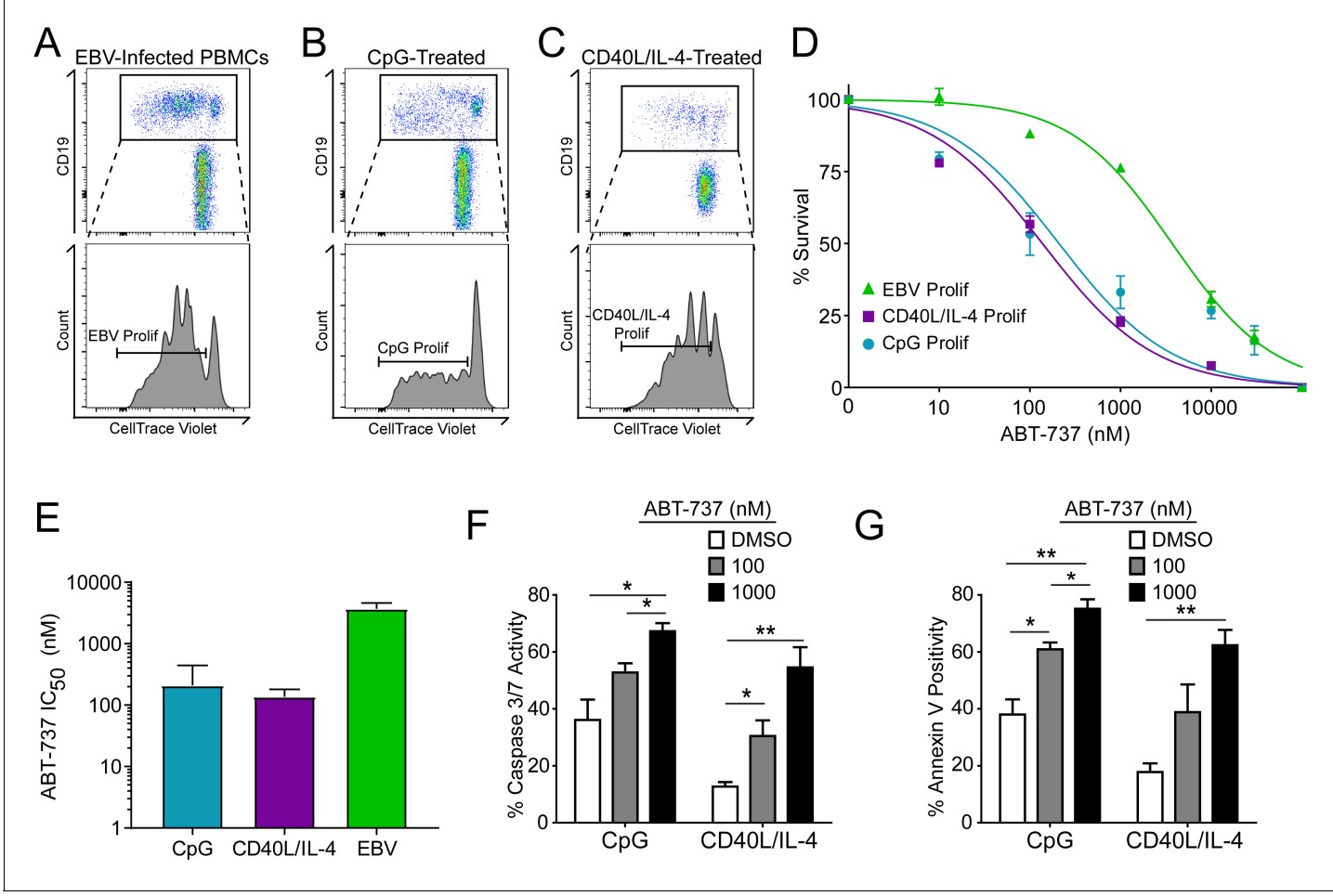

**Figure 4.** Resistance to BCL-2 antagonism is virus specific. (**A**) Flow cytometry plot of proliferating (Prolif) EBV-infected PBMCs. (**B**) Same as in (**A**), but treated with the TLR9-ligand CpG DNA. (**C**) Same as in (**A**), but treated with soluble recombinant CD40L and IL-4. (**D**) Dose-response curves generated from treating EBV-infected or mitogen-stimulated proliferating B cells with ABT-737 on Day 3.5 post infection/stimulation and reading percent survival on Day seven post infection/stimulation. Percent survival is the percent of proliferating CD19+ B cells compared to DMSO-treated controls at each time point. Data are from three human donors. (**E**) Average $IC_{50}$ with 95% Confidence Intervals are plotted for ABT-737 treatment on EBV-infected or mitogen-stimulated cells. (**F**) Caspase 3/7 activity in proliferating CpG and CD40L/IL-4 stimulated cells increases with increasing concentrations of ABT-737; values are reported as average plus SEM of three human donors. Two-tailed t-test results: CpG, DMSO vs 1000 nM (*p=0.0157); CD40L/IL-4, DMSO vs 1000 nM (**p=0.0046). (**G**) Annexin V positivity in proliferating CpG and CD40L/IL-4 stimulated cells increases with increasing concentrations of ABT-737; values are reported as average plus SEM of three human donors. Two-tailed t-test results: CpG, DMSO vs 1000 nM (**p=0.0042); CD40L/IL-4, DMSO vs 1000 nM (**p=0.0002).

The following source data is available for figure 4:

**Source data 1.** Source data for cell counts and apoptotic assays of uninfected, mitogen-stimulated B cells.

transforming primary B cells (*Miller et al., 1974*), we compared the ABT-737 sensitivity of P3HR1-infected cells induced to proliferate using the TLR9 ligand CpG (P3HR1+CpG) to uninfected B cells treated with CpG. P3HR1 infection did not alter the sensitivity of CpG-treated proliferating B cells to ABT-737 (*Figure 5B*). In contrast, the prototypical transforming strain, B95-8, induced marked ABT-737 resistance in the presence or absence of CpG (*Figure 5B*). These data implicate EBNA2 as a key determinant of the ABT-737 resistance phenotype.

Since EBNA2 controls the expression of other EBV latency genes, it was important to assess the expression of these genes during early P3HR1 and B95-8 infection. At 2 days post-infection, a time point at which ABT-737 resistance first emerges (*Figure 5—figure supplement 1*), P3HR1-infected

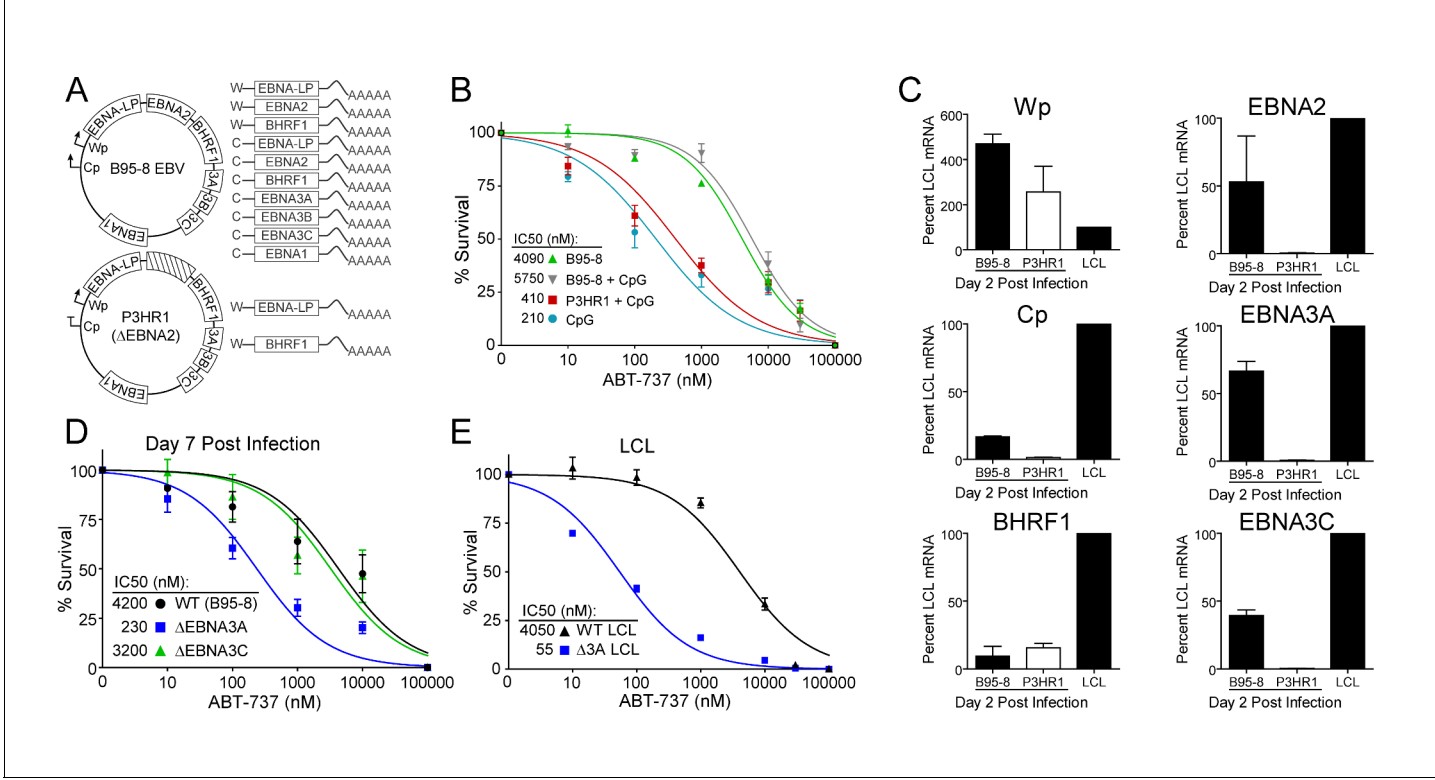

**Figure 5.** EBV-induced resistance to BCL-2 antagonism is mediated by EBNA3A. (**A**) Schematic to show the genetic differences between prototypical transforming strain of EBV (B95-8) and P3HR1 (EBNA2-deleted) strain. (**B**) Dose-response curves to assess ABT-737 sensitivity in proliferating prototypical (B95-8) EBV-infected cells and EBNA2-deleted (P3HR1) EBV-infected cells. To induce proliferation, P3HR1-infected cells required additional co-treatment with CpG DNA. Data are from three human donors. Average $IC_{50}$ is shown (inset). (**C**) qPCR showing early viral mRNAs expressed at day two post infection normalized to a B95-8 LCL. Data are shown as SEM of two matched human donors. While P3HR1 produces the same relative amount of Wp-driven transcripts and BHRF1, P3HR1 does not induce C promoter (Cp)-driven transcripts or the viral EBNA2, −3A, and −3C transcripts. (**D**) Early-infected proliferating EBNA3A-deleted (△EBNA3A) EBV-infected cells are sensitive to ABT-737, while EBNA3C-deleted (△EBNA3C) EBV-infected cells and wildtype-infected proliferating cells maintain high levels of ABT-737 resistance. Percent survival is the percent of proliferating CD19+ B cells compared to DMSO-treated controls at each time point. Data shown are from 3 to 6 human donors with SEM. Average $IC_{50}$ is shown (inset). (**E**) EBNA3A-deleted LCLs (△3A LCL) are more sensitive to ABT-737 than wildtype LCLs. Data are from three biological replicates with SEM. Average $IC_{50}$ is shown (inset).

The following source data and figure supplement are available for figure 5:

**Source data 1.** Source data for cell counts performed with cells infected with EBV mutant strains and mRNA levels of EBV viral transcripts.
**Source data 2.** Source data for *Figure 5—figure supplement 1*.
**Figure supplement 1.** ABT-737 resistance is gained within two days post infection.

cells express high levels of Wp-driven transcripts, but significantly less Cp-driven transcripts including EBNA2, EBNA3A, and EBNA3C relative to B95-8 infected cells (*Figure 5C*). As recent work indicates that the viral BCL-2 homologue, BHRF1, is expressed from Wp during primary B cell infection (*Kelly et al., 2009*) and over-expression of BHRF1 can prevent apoptosis induced by ABT-737 in mouse cells (*Kvansakul et al., 2010*), it was important to analyze BHRF1 expression. We found that P3HR1-infected cells expressed similar levels of BHRF1 as B95-8 infected cells (*Figure 5C*). Therefore, BHRF1 was not associated with ABT-737 resistance in early-infected B cells.

Resistance to ABT-737 could be conferred by EBNA2 or by a downstream target such as the EBNA3 proteins. While EBNA3A and EBNA3C are both important for B cell transformation, EBNA3A- and EBNA3C-deleted viruses can induce B cell proliferation for several weeks in culture

before aborting long-term outgrowth (*Anderton et al., 2008*; *Skalska et al., 2013*). To explore the potential role of the EBNA3 proteins in mediating ABT-737 resistance, we infected PBMCs with mutant strains of EBV deleted for EBNA3A or EBNA3C (△3A, △3C), and treated them with ABT-737 at 3.5 dpi then assayed for survival at 7 dpi as before. While △3C-infected proliferating B cells were resistant to ABT-737 (IC$_{50}$ ~3 μM) similar to WT B95-8-infected cells (IC$_{50}$ ~4 μM), the △3A-infected proliferating B cells remained highly sensitive to ABT-737 (~200 nM) (*Figure 5D*). Furthermore, from our initial infections we were able to derive a △3A LCL, similar to that reported by others (*Hertle et al., 2009*; *Skalska et al., 2010*), and we again observed enhanced sensitivity to ABT-737 and ABT-199 in the △3A LCL (IC$_{50}$ ~60 nM and ~180 nM, respectively) relative to a donor-matched WT LCL (IC$_{50}$ ~4 μM) (*Figure 5E*, *Figure 5—figure supplement 1C*). This result was unexpected as LMP1-mediated NFκB activity in a △3A LCL should induce BFL-1 to rescue ABT-737 resistance (*D'Souza et al., 2004*; *Pratt et al., 2012*) (*Figure 1E*).

## EBNA3A is required for MCL-1 mitochondrial localization and BFL-1 transcription

To further characterize the nature of EBNA3A-mediated ABT-737 resistance, we subjected △3A and WT LCLs from the same donor to BH3 profiling. We found that the △3A LCL, in contrast to WT, was sensitive to both the Bad and Bmf peptides indicating BCL-2 dependence and an absence of BFL-1 and MCL-1 protection similar to the uninfected B cell state (*Figure 6A*). Mechanistically, we found that △3A LCLs expressed significantly less BFL-1 at the mRNA level, while the BCL-2 and MCL-1 mRNA levels were similar to WT, if not slightly elevated (*Figure 6B* and additional clones in *Figure 6—figure supplement 1A*). Loss of BFL-1 mRNA expression was not due to selective pressure applied during outgrowth of △3A LCL as EBNA3A-ERT2 fusion protein expressing LCLs lost or gained BFL-1 expression upon HT withdrawal or addition, respectively, in a stable LCL (*Figure 6—figure supplement 1B–C*). While BH3-profiling predicted a loss of MCL-1 activity, MCL-1 mRNA, protein, and protein stability were unchanged in the △3A LCL compared to WT (*Figure 6B–D*). Instead, we discovered that MCL-1 localization to mitochondria was defective in the △3A LCL (*Figure 6E*). These data strongly corroborate the BH3 profiling results, which indicated a lost of MCL-1 acitivity at the mitochondria.

We further sought to determine how EBNA3A regulates BFL-1 and specifically to characterize whether the loss of BFL-1 mRNA in the △3A LCL was due to a transcriptional defect (*Figure 6B*). Indeed, when we isolated nascent mRNAs using a thiolated-RNA pull down method we found that △3A LCLs were only transcribing ~15% as much BFL-1 mRNA as WT LCLs (*Figure 6F*). Furthermore, we found that decreased BFL-1 transcription in △3A LCLs was correlated with decreased RNA polymerase II (Pol II), activated Pol II (phospho Serine 5), and activating histone marks (H3K27ac, H3K9ac, and H3K4me3) accumulating at the BFL-1 transcription start site (TSS) (*Figure 6G*).

EBNA3 proteins often regulate gene expression through long-range enhancers (*Bazot et al., 2015*; *McClellan et al., 2013*; *Schmidt et al., 2015*). To evaluate the role of EBNA3A in regulating chromatin structure to promote BFL-1 transcription, we used chromatin conformation capture to query the interactions between regions of open chromatin up- and downstream of the BFL-1 gene in WT and △3A LCLs. We discovered two genomic regions upstream of the BFL-1 gene that were strongly associated with the TSS in WT LCLs but less so in △3A LCLs (*Figure 6H*). Using LCL ChIP-Seq data (*Schmidt et al., 2015*; *Zhao et al., 2011*), we found that one locus is bound by EBNA3A within a known EBV Super Enhancer (39 kb upstream of the BFL-1 TSS) and the other region contains a single RelA peak, an NFκB transcription factor subunit strongly activated by LMP1 (*Zhao et al., 2014*) (*Figure 6I*). These data support an important new role for EBNA3A in coordinating NFκB-mediated regulation of BFL-1 transcription in LCLs.

Finally, to demonstrate that EBNA3A is responsible for ABT-737 resistance in EBV-infected B cells, we rescued EBNA3A expression in trans in △3A LCLs. We found that EBNA3A re-expression was sufficient to restore ABT-737 resistance similar to WT LCLs (*Figure 6J–K*). Therefore, we conclude that EBNA3A plays an important role in EBV-mediated resistance to BCL-2 antagonists and survival in response to potent proliferative signals during both early and late stages of B cell growth transformation.

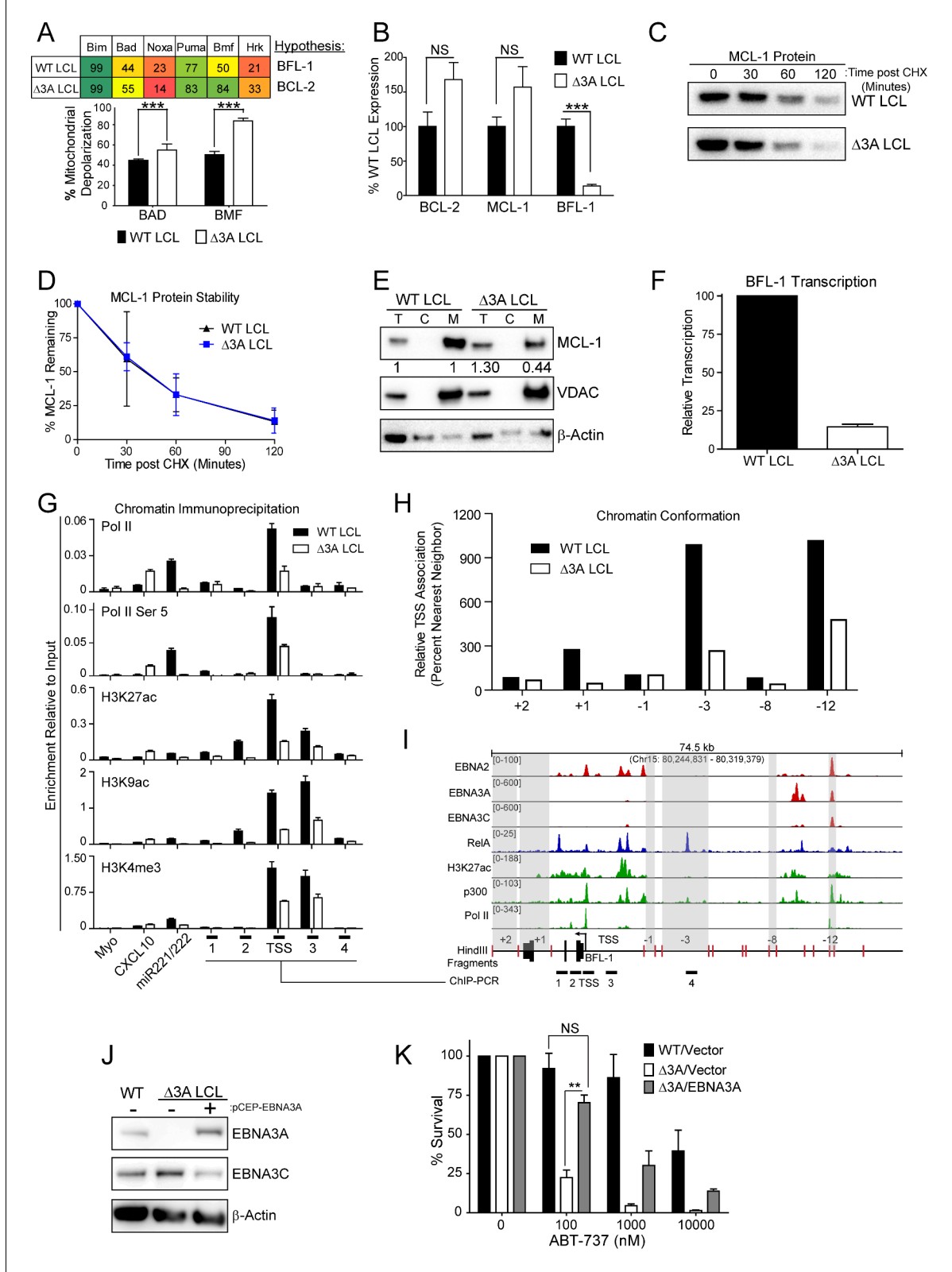

**Figure 6.** EBV EBNA3A is required for MCL-1 mitochondrial localization and BFL-1 transcription. (**A**) BH3 profile shows increased sensitivity to Bad and Bmf peptides in EBNA3A-deleted LCLs compared to wildtype LCLs, indicative of a BCL-2 dependence. Three technical replicates with seven repeated measures each over time were averaged and plotted with SEM. Data were analyzed by paired 2-tailed t-test, *p<0.001. (**B**) qPCR of BCL-2, MCL-1, and BFL-1 mRNA levels in matched wild type and EBNA3A-deleted LCLs from four human donors. NS, not significant, ***p=0.002 (**C**) Western blot of MCL-
*Figure 6 continued on next page*

*Figure 6 continued*

1 in WT or △3A LCLs treated with 15 µM cycloheximide (CHX) over two hours. (D) The relative mean densitometry from panel (C) plotted over time from two biological replicates plus SEM. (E) Sub-cellular fractionation into total (T), cytoplasmic (C), or mcitochondrial (M) compartments reveals MCL-1 mislocalization in an EBNA3A-deleted LCL compared to a Wildtype LCL. Immunoblot for MCL-1, VDAC (mitochondrial localization control), and β-Actin (total lysate control). Quantified levels of MCL-1 in total and mitochondrial compartments are normalized to their respective WT levels and VDAC control. (F) Relative transcription rate of BFL-1 in WT and △3A LCLs ascertained by qPCR on pulldown of nascent mRNAs. Data is shown as the average and SEM of three biological replicates. (G) Chromatin immunoprecipitation (ChIP) was performed on extracts from WT and △3A LCLs using antibodies for total RNA Pol II (Pol II), Pol II phospho-Ser 5 (Pol II Ser 5), H3K27ac, H3K9ac, and H3K4me3. Primer pairs for Myo were used as a negative control, CXCL10 as an EBNA3A-repressed control, and miR221/222 as an EBNA3A-activated control. Primer pairs surrounding the BFL-1 TSS are shown annotated in panel (I) in their proper locations. Values represent ratio of chromatin precipitated, after correction for IgG, relative to 2.5% of input. Data are shown as the mean and standard deviation (SD). (H) Chromatin conformation capture (CCC) was performed on WT and △3A LCLs, and the BFL-1 loci were interrogated for interaction after digesting with HindIII. Relative interaction frequency to the BFL-1 TSS fragment (panel I bottom) was assayed by qPCR and normalized to the interaction frequency of the nearest neighbor (−1) fragment set at 100% relative interaction. Results are the average of two independent experiments. (I) ChIP-Seq data for EBNA2, EBNA3A, EBNA3C, NFκB (RelA), Histone H3K27ac, p300, and RNA Pol II (Pol II) from an LCL on the *BCL2A1* (BFL-1) locus. (J) EBNA3A protein expression can be rescued to wild-type LCL levels in a △3A LCL using an episomal EBNA3A expression vector. (K) Rescuing EBNA3A expression in △3A LCLs restores resistance to ABT-737. Three individual clones of WT, △3A, or △3A/EBNA3A were subjected to ABT-737 treatment for three days and then analyzed for viability by FACS. Remaining viable cells on day three post treatment are normalized to the untreated (0 nM) cells. Data were analyzed by 2-way ANOVA and showed that there was a significant interaction (p<0.001) between cell type and drug treatment. NS, not significant; **p=0.01 by two-tailed t-test.

The following source data and figure supplement are available for figure 6:

**Source data 1.** Source data for individual responses to BH3 peptides, mRNA and protein levels.
**Figure supplement 1.** Additional validation of BFL-1 mRNA regulation by EBNA3A.

## Discussion

Suppression of apoptosis by EBV is essential for establishment of latent infection in the immune-competent host and promotes tumorigenesis in the immune suppressed host. While EBV encodes two putative BCL-2 homologues, these proteins are not required for survival after the first 24 hr following B cell infection (*Altmann and Hammerschmidt, 2005*). Rather, constitutive NFκB signaling from the viral oncoprotein LMP1, a TNFR homologue, is thought to be responsible for apoptotic suppression in latently infected, proliferating cells (*Cahir-McFarland et al., 2000*; *Pratt et al., 2012*). However, the initial burst of B cell proliferation induced by EBV is driven by expression of the viral EBNA transcription factors (EBNA1, 2, 3A, 3B, 3C, and LP) in the absence of appreciable LMP1 or NFκB activity (*Price et al., 2012*). Furthermore, this early period of hyper-proliferation activates a growth-suppressive DNA damage response, which would be expected to promote apoptosis (*Nikitin et al., 2010*). We, therefore, sought to identify the mechanism of survival during early infection by using BH3 profiling to query mitochondrial control of apoptosis prior to and through latent EBV infection and outgrowth of primary human B cells into LCLs. While BH3 profiling has been previously performed on virus-infected cells (*Cojohari et al., 2015*), this study focused on HCMV and was limited to a single EBV-positive Burkitt lymphoma cell line, Akata, which expresses only the viral EBNA1 protein. In our BH3 profiling study, we found that mature human peripheral blood B cells depend solely on BCL-2 for survival while EBV latent infection further suppressed apoptotic priming during outgrowth by initially upregulating MCL-1, followed by BFL-1. Mechanistically, we identified a new and unique role for the viral latent protein EBNA3A in apoptosis resistance during both early and late EBV-induced B cell proliferation.

The importance of EBNA3A in B cell transformation and survival was previously recognized, but our studies now define the molecular basis for the inhibition of apoptosis. While both EBNA3A and EBNA3C suppress the pro-apoptotic protein Bim (*Paschos et al., 2009*), only EBNA3A was required to mitigate resistance to a BCL2 antagonist through outgrowth. EBNA3A regulated the BH3 profile of EBV-infected cells and suppressed apoptosis by promoting MCL-1 mitochondrial localization and increasing BFL-1 mRNA transcription. Transcription of BFL-1 was previously shown to be regulated by the EBV transcription factor EBNA2 (*Pegman et al., 2006*) as well as LMP1-induced NFκB (*Pratt et al., 2012*). Our work now indicates that EBNA3A coordinates a chromatin architecture through an EBV super enhancer that facilitates NFκB-mediated activation of BFL-1 transcription.

Therefore, our studies highlight a new cooperative role among EBNA3A, EBNA2, and LMP1 in transcriptional activation important for EBV-immortalized cell survival.

The importance of MCL-1 in early EBV-infected human B cells coupled with the critical role of MCL-1 in germinal center (GC) formation and memory B cell survival in the mouse (*Vikstrom et al., 2010*) is consistent with the GC model of EBV infection. The GC model posits that EBV infection of naïve B cells promotes B cell maturation mimicking the GC reaction to access the long-lived memory B cell compartment and thus persisting for the lifetime of the host (*Burns et al., 2015*; *Roughan and Thorley-Lawson, 2009*). Our data support this model in that early infection of B cells by EBV promotes MCL-1 as a critical apoptotic regulator similar that observed for GC B cells in the mouse (*Vikstrom et al., 2010*). Furthermore, MCL-1 copy number gains and mutations are often found in B cell lymphomas (*Wenzel et al., 2013*). Therefore, the mechanism by which EBNA3A enhances MCL-1 function at the mitochondria may provide broad insight into apoptotic regulation during B-cell maturation and lymphomagenesis.

The early phase of EBV infection where EBNAs are expressed with little or no expression of LMPs is known as latency IIb (*Price and Luftig, 2015*). The latency IIb expression pattern has also been observed by immuno-histochemical analysis in EBV-positive HIV-associated lymphomas, post-transplant lymphomas, in acute infectious mononucleosis, and in tumors of humanized mouse models of EBV infection (*Price and Luftig, 2015*). Indeed, a recent study found that an EBV recombinant lacking LMP1 was capable of forming B cell tumors in a humanized mouse model (*Ma et al., 2015*). The survival of these cells as well as those in other latency IIb expressing tissues likely depends on EBNA3A and MCL-1, while latency III cells expressing LMP proteins and high level NFκB signaling are likely to be less primed and to rely on BFL-1. As BH3 profiling has been a powerful predictive tool for chemotherapeutic responses in leukemias (*Montero et al., 2015*), our new findings suggest that such an approach could allow for better treatment decisions for EBV-associated malignancies.

## Materials and methods

### Cells, viruses, and mitogens

Buffy coats were obtained from normal human donors through the Gulf Coast Regional Blood Center (Houston, TX) and peripheral blood mononuclear cells (PBMCs) were isolated by Ficoll Histopaque-1077 gradient (Sigma, H8889). CD19+ B cells were purified from PBMCs using the BD iMag Negative Isolation Kit (BD, 558007). Lymphoblastoid cell lines were generated as previously described (*Price et al., 2012*) and maintained in RPMI plus 10% FBS (Corning), 2 mM L-Glutamine, 100 U/ml penicillin, 100 μg/ml streptomycin (Invitrogen).

B95-8 strain of Epstein-Barr virus was produced from the B95-8 Z-HT cell line as previously described (*Johannsen et al., 2004*). P3HR1 was generated from P3HR1 Z-HT cells (kind gift of Elliott Kieff, Harvard Medical School). Two independent sources of EBNA3A and EBNA3C deletion viruses were used in these studies for validation as well as an EBNA3A-ERT2 regulated recombinant (*Bazot et al., 2015*). Detailed information on mutant viruses below.

TLR9 ligand CpG oligonucleotide (ODN 2006) was purchased from IDT and used at 2.5 μg/ml (*Krieg et al., 1995*). Human recombinant interleukin-4 (PeproTech, AF200-04) was used at 20 ng/mL. HA-tagged CD40 ligand was purchased from (R&D Systems, 6420 CL) and used at 5 ng/ml in combination with an anti-HA cross-linking peptide (R&D Systems, MAB060; RRID:AB_10719128) at a concentration of 0.2 μg/μl.

Initial BH3 profiling experiments were performed with cell populations and lines that were not mycoplasma tested. However, subsequently we instituted routine mycoplasmia testing using a PCR-detection kit (Sigma-Aldrich, MP0035-1KT). While the majority of cell lines were negative for mycoplasma, we did detect mycoplasma in some lines. These lines were cured with ciprofloxacin treatment. We repeated several key experiments in those lines that were found to be positive and then cured, including ABT-737 and ABT-199 sensitivity assays, and found that mycoplasma contamination had no bearing on the results.

### Virus strain generation and production

Two independent sources of EBNA3A and EBNA3C deletion viruses were used in these studies for validation. One set was previously described (*Anderton et al., 2008*). The other was generated by

en passant mutagenesis of the EBV p2089 BACmid (*Delecluse et al., 1998*) using GS1783 *E. coli* (*Braman, 2010*). Primer sequences used to generate these mutants are listed in *Supplementary file 1*. After screening, desired EBV BACmids were transferred to BM2710 *E. coli* which were then used to infect 293 cells as previously described (*Chen et al., 2005*).

Recombinant viruses were produced by co-transfecting BAC-containing, selected 293 cells with pCDNA3-2670-gp110, pSG5-Zta, and pCDNA3-Rta into cells using polyethylenimine (PEI) and Opti-MEM (Invitrogen). 24 hr after transfection, media was changed to antibiotic-free 10% FBS containing RPMI 1640 and harvested 48 hr afterwards. After harvesting, transfected cells were replenished with fresh media for 24 hr, at which point virus was harvested again. Harvests were pooled and concentrated 100-fold in a 100 kDa Amicon spin-filter.In addition to the EBNA3A deletion and stop variants, a recombinant virus was used that encodes EBNA3A-ERT2, which is a fusion of a modified estrogen receptor to EBNA3A. This virus was previously used to define EBNA3A regulated genes and its generation was described here (*Bazot et al., 2015*).

## Virus infections

EBV infections were performed by adding 100 µL B95-8 virus (from Z-HT supernatants) or 200 µL P3HR1 virus to $1 \times 10^6$ PBMCs for 1 hr at 37°C in a $CO_2$ incubator followed by washing the cells once with PBS and resuspending cells in RPMI with 15% FBS serum (Corning), 2 mM L-Glutamine, 100 U/ml penicillin, 100 µg/ml streptomycin (Invitrogen), and 0.5 µg/mL Cyclosporine A (Sigma). Cells were cultured at 37°C in a humidified incubator at 5% $CO_2$ (*Nikitin et al., 2010*). For P3HR1 infections, viral stocks were normalized to B95-8 by immunofluorescence of EBNA-LP in primary B cells two days post infection. For WT, △3A, and △3C BAC viruses, $1 \times 10^5$ green Raji units (GRU; Raji, RRID:CVCL_2699) were used to infect $1 \times 10^6$ PBMCs as previously described (*Skalska et al., 2013*).

## Generation of CRISPR/Cas9 mutants

The BFL-1 CRISPR/Cas9 mutant LCL was generated by the Duke Functional Genomics Shared Resource. Cas9-expressing LCLs (lentiCas9-EGFP; Addgene #63592) were transduced with one of three short-guide RNAs (sgRNAs) that were specific to BFL-1 exon 1 and grown out under puromycin selection (lentiGuide-Puro; Addgene #52963). Each sgRNA was screened for efficiency of Cas9 cleavage by Surveyor nuclease assay and for reduced BFL-1 mRNA expression. LCLs transduced with the sgRNA resulting in both Cas9 cleavage of the target site and the most reduced mRNA expression were then serially diluted to obtain a clonal population. Clonality was ascertained by TOPO TA cloning (Invitrogen, Cat #45003).

## Flow cytometry and sorting

To track proliferation, cells were stained with CellTrace Violet (Invitrogen, C34557). Cells were first washed in FACS buffer (5% FBS in PBS), stained with the appropriate antibody for 30 min-1hr at 4°C in the dark, and then washed again before being analyzed on a BD FACS Canto II.

Proliferating infected B cells were sorted to a pure population of CD19+/CellTraceViolet^lo on a MoFlo Astrios Cell Sorter at the Duke Cancer Institute Flow Cytometry Shared Resource. Mouse anti-human CD19 antibody (clone 33-6-6; gift from Tom Tedder, Duke University Medical School) conjugated with either APC or PE was used as a surface B-cell marker in flow cytometry.

## BH3 profiling (JC-1 plate-based assay)

Plate-based BH3 profiling was performed on primary B cells, early-infected hyperproliferating B cells, and LCLs at $5 \times 10^4$ cells/well. Cells were resuspended at 2x and incubated for 10 min at room temperature (RT) in T-EB solution containing 4 µM JC-1, 40 ug/mL oligomycin, 0.02% digitonin, and 20 mM 2-mercaptoethonal. Peptides were diluted in T-EB at 2x concentration and aliquoted per well in a black 384-well plate (Corning 3575), to which the cell/dye solution was added 1:1. The fluorescence at 590 nM from each well was read on a plate reader every 5 min at RT. Depolarization of the mitochondria was calculated as a percentage loss normalized to the solvent-only control (DMSO, 0% depolarization) and the positive control (FCCP, 100% depolarization). Analysis was performed using technical triplicates of five biological replicates, where outliers were eliminated if there was poor signal or if the Puma2A inert control showed high background depolarization.

## Protein expression

Cells were pelleted and washed in PBS, and then lysed in 0.1% Triton-containing buffer with Complete protease inhibitors. All protein lysates were run on NuPage 4–12% gradient gels (LifeTechnology) and transferred to PVDF membrane (GE Healthcare). Membranes were blocked in 5% milk in TBST and stained with primary antibody overnight at +4°C, followed by a wash and staining with secondary HRP-conjugated antibody for 1 hr at room temperature. All antibodies used can be found in the *Supplementary file 1*.

## Gene expression analysis

Total RNA was isolated from cells by using a Qiagen RNeasy kit and then reverse transcribed to generate cDNA with the High Capacity cDNA kit (Applied Biosystems). Quantitative PCR was performed by using SYBR green (Quanta Biosciences) in an Applied Biosystems Step One Plus instrument. Results were normalized to SETDB1, a control mRNA found to not change expression from resting B cells through EBV-immortalization (*Price et al., 2012*) or GNB2LI (*Bazot et al., 2015*).

## Small molecule inhibitors

Cells were treated with various concentrations of ABT-199 (ChemieTek, CT-A199), ABT-737 (Selleckchem, S1002), flavopiridol (Sigma, F3055), or A1210477 (Selleckchem, S7790). All drugs were resuspended in DMSO.

## Cell viability assays

Cells were stained with fluorescent conjugates of Annexin-V (eBioscience, 88-8007-72; RRID:AB_2575165) or incubated with CellEvent Caspase 3/7 Green Detection Reagent (ThermoFisher Scientific, C10423). Cell death is reported as a percentage of Annexin-V positive or caspase 3/7-positive of total events. Survival is plotted as the percentage of CD19+, proliferating B cells normalized by input volume using the FACS Canto II high-throughput 96-well sampler to DMSO-treated control cells.

## Protein stability

Protein stability was assayed by incubating LCLs in 15 μM cycloheximide (Sigma, C1988). Time points were taken immediately before addition (0 min), and then at 30, 60, and 120 min after addition of cycloheximide and analyzed by western blot loading 10 μg of protein per lane.

## Mitochondrial isolation

Mitochondria isolation was performed using a mitochondria isolation kit (Thermo Scientific) on 50 million cells. Cells were lysed with Reagents A, B, and C as supplied by the manufacturer's kit. Mitochondrial pellets were lysed in 0.1% Triton lysis buffer with protease and phosphatase inhibitors. Total, cytoplasmic, and mitochondrial fractions were measured by Bradford assay and 5 μg protein was loaded per lane.

## Nascent transcription profiling

The ability to capture nascently transcribed RNA was modified from previously described work (*Payne et al., 2014*). In brief, cells were treated with 200 μM 4-thiouridine (4sU) for one hour before total RNA was harvested in 1 mL of TRIzol (Life Technologies). Nascently transcribed RNA over this time period incorporated 4sU, which was subsequently biotinylated and pulled down using streptavidin MyOne C1 Dynabeads (Invitrogen). RNA was reverse transcribed and qPCR was performed as described above.

## Chromatin immunoprecipitation

Chromatin Immunoprecipitation (ChIP) was performed as described previously (*Bazot et al., 2015*) using a Millipore ChIP assay kit (17-295) and sonicated with a standard Diagenode Bioruptor sonicator. All antibodies used can be found in the *Supplementary file 1*.

## Chromatin conformation capture

Chromatin Conformation Capture (CCC) was performed as described previously (*Hagège et al., 2007*). In brief, chromatin was crosslinked with formaldehyde and isolated from LCLs, whereupon it was digested overnight with the restriction enzyme HindIII. Digested chromatin was then diluted and ligated overnight. Reversal of crosslinks, proteinase K digestion, and phenol-chloroform extraction yielded DNA circles that were assayed by TaqMan qPCR. All oligos and probes used can be found in the *Supplementary file 1*.

## Transfection of LCLs

pCEP-EBNA3A, an episomal expression vector, was introduced into LCLs using the MicroPorator MP-100 (Digital Bio). Cells were incubated with DNA and shocked at 1300V for 1 pulse of 30 ms and then recovered in 15% FBS containing RPMI. Cells were co-transfected with an additional pCEP-TdTomato plasmid to mark transfected cells with red fluorescence.

## Statistical analysis

Student's 2-way ANOVA and two-tailed t-test were calculated using GraphPad Prism 5.0 software. *$p<0.05$, **$p<0.01$, and ***$p<0.001$ were considered significant. $IC_{50}$ values were calculated by non-linear regression using GraphPad and 95% confidence intervals were reported.

# Acknowledgements

We thank Lynn Martinek, Nancy Martin, and Mike Cook for extensive help in flow-based cytometry experiments and Karyn McFadden for critical reading of the manuscript. Special thanks are due to Zach Carico for help in optimizing CCC experimental design. So-Young Kim, the Director of the Duke Functional Genomics Shared Resource was very helpful in designing and discussing BFL-1 sgRNA targeting vectors. This work was supported by National Institutes of Health (NIH) Grants R01-CA140337 and R01-DE025994 (to MAL), R01-DE023939 (to EJ), R01-CA129974 (to AL), F31-CA180451 (to AMP), T32-CA009111 (to AMP and JD), and T32-AI078985 (to RD). Additional funding came from the Duke CFAR, an NIH-funded program, 5P30-AI064518, and an American Cancer Society grant RSG-13-228-01-MPC (Both to MAL). Funding for QB and MJA was provided by the Wellcome Trust (099273/Z/12/Z). We would like to dedicate this manuscript to the memory of Martin J Allday, a generous colleague and leader in the field of EBV biology.

# Additional information

### Competing interests

AL: Is a paid advisor to, and his laboratory receives research sponsorship from, AbbVie, Astra-Zeneca, and Tetralogic. The other authors declare that no competing interests exist.

### Funding

| Funder | Grant reference number | Author |
|---|---|---|
| National Cancer Institute | F31-CA180451 | Alexander M Price |
| National Institutes of Health | T32-CA009111 | Alexander M Price<br>Joanne Dai |
| National Institute for Dental and Craniofacial Research | R01-DE025994 | Joanne Dai<br>Micah A Luftig |
| Wellcome | 099273/Z/12/Z | Quentin Bazot<br>Martin J Allday |
| National Institute for Allergy and Infectious Diseases | T32-AI078985 | Reza Djavadian |
| National Institute for Dental and Craniofacial Research | R01-DE023939 | Eric C Johannsen |
| National Cancer Institute | R01-CA129974 | Anthony Letai |
| National Cancer Institute | R01-CA140337 | Micah A Luftig |

| American Cancer Society | RSG-13-228-01-MPC | Micah A Luftig |
| National Institute for Allergy and Infectious Diseases | 5P30-AI064518 | Micah A Luftig |

The funders had no role in study design, data collection and interpretation, or the decision to submit the work for publication.

## Author contributions

AMP, Conceptualization, Data curation, Formal analysis, Validation, Investigation, Methodology, Writing—original draft, Writing—review and editing; JD, Data curation; QB, Formal analysis, Validation, Investigation; LP, PSW, Formal analysis, Investigation; PAN, Conceptualization, Data curation, Investigation, Writing—review and editing; RD, Investigation, Writing—review and editing; CAS, Investigation, Performed experiments related to the sensitivity of early-infected B cells to ABT-199 at different times of addition, Analyzed the data together with AMP and MAL; APB, Investigation, Methodology; KCW, Resources, Supervision, Writing—review and editing; ECJ, Supervision, Writing—original draft, Writing—review and editing; AL, Resources, Data curation, Supervision, Funding acquisition, Methodology, Writing—original draft; MJA, Resources, Formal analysis, Supervision, Funding acquisition, Writing—original draft; MAL, Conceptualization, Data curation, Supervision, Funding acquisition, Visualization, Writing—original draft, Project administration, Writing—review and editing

## Author ORCIDs

Alexander M Price, http://orcid.org/0000-0003-4654-4604
Joanne Dai, http://orcid.org/0000-0002-9879-4704
Micah A Luftig, http://orcid.org/0000-0002-2964-1907

# Additional files

## Supplementary files

• Supplementary file 1. Antibodies used for western blot and chromatin immunoprecipitation are included below.

## Major datasets

The following previously published datasets were used:

| Author(s) | Year | Dataset title | Dataset URL | Database, license, and accessibility information |
|---|---|---|---|---|
| Zhao B, Zou JY, Wang H, Johannsen E, Aster J, Bernstein B, Kieff E | 2011 | EBNA2 ChIP-Seq | https://www.ncbi.nlm.nih.gov/geo/query/acc.cgi?acc=GSE29498 | Publicly available at the NCBI Gene Expression Omnibus (accession no: GSE29498) |
| Zhao B, Barrera LA, Ersing I, Willox B, Schmidt S, Zhou H, Shi T, Mollo S, Greenfeld H, Takasaki K, Jiang S, Cahir-McFarland E, Kellis M, Bulyk ML, Kieff E, Gewurz B | 2014 | The NF-kB genomic landscape in lymphoblastoid B-cells | https://www.ncbi.nlm.nih.gov/geo/query/acc.cgi?acc=GSE55105 | Publicly available at the NCBI Gene Expression Omnibus (accession no: GSE55105) |
| Jiang S, Willox B, Zhou H, Holthaus A, Wang A, Shi T, Maruo S, Johannsen E, Kharchenko P, Kieff E, Zhao B | 2013 | EBNA3C ChIP-Seq | https://www.ncbi.nlm.nih.gov/geo/query/acc.cgi?acc=GSE52632 | Publicly available at the NCBI Gene Expression Omnibus (accession no: GSE52632) |
| Schmidt SC, Jiang S, Willox B, Zhou H, Holthaus AM, | 2015 | EBNA3A ChIP-Seq | https://www.ncbi.nlm.nih.gov/geo/query/acc.cgi?acc=GSE59181 | Publicly available at the NCBI Gene Expression Omnibus |

| | | | | |
|---|---|---|---|---|
| Johannsen E, Kharchenko P, Zhao B, Kieff E | | | | (accession no: GSE59181) |
| Shoresh N | 2011 | Histone modifications in LCLs (ENCODE) | https://www.ncbi.nlm. nih.gov/geo/query/acc. cgi?acc=GSE29611 | Publicly available at the NCBI Gene Expression Omnibus (accession no: GSE29611) |
| Snyder M, Gerstein M, Weissman S, Farnham P, Struhl K | 2011 | TF binding sites in LCLs (ENCODE) | https://www.ncbi.nlm. nih.gov/geo/query/acc. cgi?acc=GSE31477 | Publicly available at the NCBI Gene Expression Omnibus (accession no: GSE31477) |

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
