## [Decision Letter]

Thank you for submitting your article "Epstein-Barr virus ensures B-cell survival by uniquely modulating apoptosis at early and late times after infection" for consideration by *eLife*. Your article has been favorably evaluated by Michel Nussenzweig (Senior Editor) and three reviewers, one of whom is a member of our Board of Reviewing Editors. The reviewers have opted to remain anonymous.

The reviewers have discussed the reviews with one another and the Reviewing Editor has drafted this decision to help you focus on concerns raised by the reviews. Given the nature of some of these concerns, we request that you respond with a plan of experiments to address the essential revisions and a time table for their completion. The editor and reviewers will consider your response and issue a recommendation.

Summary:

This paper identifies a mode of action of the EBV EBNA3A gene product in preventing apoptosis early in B cell infection, before the better-known block to apoptosis caused by LMP1 later in infection via NFkB activation. It is further suggested that EBNA3A acts by upregulating MCL-1 and BFL-1, two anti-apoptotic factors.

There seems to be truly novel information here, about EBV and EBNA3A in particular. Previous work has shown that EBNA3A is a trans-acting gene regulator, most often a negative regulator, but not so much is known about its mode of action or its targets – it has been shown to downregulate Bim and some other genes. Here the proposal is that it upregulates two genes, MCL-1 and BFL-1. This seems novel and interesting. The MCL-1 increase (Figure 3) makes sense but no early increase in BFL-1 was observed. EBNA3A also affects MCL-1 mitochondrial localization.

Much of the work depends on antiapoptotic drugs whose specificity is assumed. The shifts in IC_50_ look impressive. Pairs of antiapoptotic drugs seem to confirm the predictions. (The use of flavopiridol to stall NRA Pol II, and show loss of MCL-1 is OK but surely a host of unstable proteins would also be affected.) The resistance to the drugs seems to be mediated by EBNA2, and then EBNA3A, based on relevant EBV mutants – this seems to be the key experiment. The effects of EBV seem distinctive from those seen with other antiapoptotic mediators.

In addition to key experiments listed below, some rewriting will be necessary to make the story more accessible and significant.

Essential revisions:

It would help nonvirologists if we had a little more background about the system and the assays and drugs.

The entrée into the work was "BH3 profiling", a way to survey peptides as probes for apoptotic factors. This method is not well described in the paper – we need more introduction to this method (Introduction, last paragraph). Please explain more about what the results can tell us (i.e. tell us about implications of a given peptide effect on mitochondrial depolarization). We run into incomprehensibility quickly (subsection “MCL-1 and BCL-2 protect EBV-infected proliferating B cells from apoptosis early after infection, while BFL-1 additionally protects LCLs from apoptosis late in infection”, first paragraph and Figure 1).

Although the authors studied three cell states, which are different from the previous work using BH3 profiling (Olesea Cojohari et al., JVI, 2015), the manuscript has already lost some of its attractiveness and significance due to the limited level of original creativity as well as weakly designed controls and experimental design. Specific comments for the authors' consideration are listed below.

1) Figure 1. Although the selective interactions in BH3 profiles suggest the potential functions of BCl-2 family member on EBV-induced B-cell proliferation, the analysis of mitochondrial depolarization in these experiments clearly shows that EBV infection plays a limited functional role in mitochondrial priming, especially in 10µm Puma or 1µm Bim treated LCLs. Meanwhile, the statistics analysis is also missing in this figure. So, these statements are a bit overreaching and should be toned down to match the data.

2) Both BCl-2 and BCl-w have similar characteristics in the BH3 profiles (Figure 1), but why do the authors only focus on the function of BCl-2 in the following experiments? For instance, the average IC_50_ for "Prolif" and "LCL" are completely opposite after treatment with ABT-737 or ABT-199, suggesting unique functions of BCl-xL and BCl-w in these cells (Figure 2). However, only BCl-2 expression was detected in Figure 3. Furthermore, in contrast to WT LCL, Δ3A LCL was sensitive to both Bad and Bmf peptides (Figure 6). Why does this indicate BCl-2 dependence, but not BCl-w dependence?

3) EBV could induce potent resistance to ABT-199, a specific inhibitor of BCL-2, so it would be better to investigate EBV-induced resistance to BCl-2 antagonism with ATB-199, instead of ABT-737. However, of particular note is that the use of the inhibitors can be misleading and cannot be totally substantiated by the results and the conclusion made without further validation.

4) The expression change of MCL-1 is different from BCl-2/Bfl-1's (Figure 3). Therefore, it seems that MCL-1 and BCl-2/Bfl-1 protect EBV-infected B-cells from apoptosis early after infection and late in infection, respectively. As shown in Figure 1, the survival mechanism may also rely on BCl-2/BCl-w to block Noxa-induced depolarization in LCLs. The authors also noted there was no available BCl-1 antibody (subsection “MCL-1 and BCL-2 protect EBV-infected proliferating B cells from apoptosis early after infection, while BFL-1 additionally protects LCLs from apoptosis late in infection”, first paragraph), actually, Bfl-1 is also known as BCL2A1 (Bcl2 related protein A1), so there should be tons of commercial antibodies at least for Bfl-1 as well as BCl-2. Therefore, Bfl-1 expression should be detected in these studies and is important to support the conclusions. In addition, the Annexin V positivity of ABT-737-treated LCLs in Figure 2 and Figure 3 are inconsistent, so this conclusion is also difficult to accept and is not convincing.

5) The Caspase 3/7 activity in ABT-737-treated "Prolif" B-cells increases according to the statistical analysis (Figure 2). Why do the authors indicate that they do not in Figure 4 (subsection “Resistance to BCL-2 antagonism is virus specific”)? The Annexin V positivity should increase in "Prolif" B-cells following ABT-737 treatment (Figure 2). It's better to mark t-test results in the figures, but not in separate files.

6) Figure 5. Are the qPCR results of viral mRNA normalized to B95.8 LCL as the authors indicated? Why are "B-cell" always shown in the figures? And what do the authors mean by "B-cell" in this manuscript? This is not clear. Is it transformed B-cell, proliferation B-cell Burkitt's lymphoma B-cells?

7) In Figure 6, Δ3A LCL expresses more BCl-2 and MCl-1 than WT at the mRNA level, which is different from the statement "similar to WT" (subsection “EBNA3A is required for MCL-1 mitochondrial localization and BFL-1 transcription”, first paragraph) based on your 2-way ANOVA analysis. The loading control is also missing in Figure 6, while the samples are not normalized in Figure 6. Without this information or control data, it's difficult to evaluate this section of the figure.

8) If decreased Bfl-1 transcription inΔ3A LCL is related with Bfl-1 TSS (Figure 6), have the authors determined the transcription activity of Bfl-1 promoter? How about the Bfl-1 protein expression inΔ3A LCL?

---

## [Author Response]

Essential revisions:

It would help nonvirologists if we had a little more background about the system and the assays and drugs.

The entrée into the work was "BH3 profiling", a way to survey peptides as probes for apoptotic factors. This method is not well described in the paper – we need more introduction to this method (Introduction, last paragraph). Please explain more about what the results can tell us (i.e. tell us about implications of a given peptide effect on mitochondrial depolarization). We run into incomprehensibility quickly (subsection “MCL-1 and BCL-2 protect EBV-infected proliferating B cells from apoptosis early after infection, while BFL-1 additionally protects LCLs from apoptosis late in infection”, first paragraph and Figure 1).

We will revise the manuscript to include a more comprehensible overview of BH3 profiling. In particular, we will address what the BH3 profile can tell us in terms of hypotheses generated regarding the BCL2 family members important for protection from apoptosis in the cell states being tested. We will consult with non-specialists to improve the clarity of the manuscript.

Although the authors studied three cell states, which are different from the previous work using BH3 profiling (Olesea Cojohari et al., JVI, 2015), the manuscript has already lost some of its attractiveness and significance due to the limited level of original creativity as well as weakly designed controls and experimental design.

The Olesea Cojohari et al., paper does use BH3 profiling, a technique developed by the Letai lab to query intrinsic mitochondrial apoptosis regulation. However, with regard to EBV, the only infected cells used are Latency I Akata Burkitt’s lymphoma cells. These cells only express the viral EBNA1 protein and miRNAs, they contain an Ig/c-Myc translocation, p53 mutation, and many other mutations that have facilitated tumor development. In contrast, our work uses primary B cells from normal donors and we infect with EBV and allow the latency program to play out (first two weeks EBNAs only, then LCLs with latency III or EBNAs and LMPs). The major question that we are addressing is how EBV is able to suppress apoptosis in this initial latency phase in the absence of the canonical survival factor, LMP1.

Specific comments for the authors' consideration are listed below.

1) Figure 1. Although the selective interactions in BH3 profiles suggest the potential functions of BCl-2 family member on EBV-induced B-cell proliferation, the analysis of mitochondrial depolarization in these experiments clearly shows that EBV infection plays a limited functional role in mitochondrial priming, especially in 10µm Puma or 1µm Bim treated LCLs. Meanwhile, the statistics analysis is also missing in this figure. So, these statements are a bit overreaching and should be toned down to match the data.

To address the high variance among the biological replicates, we performed paired t-tests on Bad and Bmf peptide treatments, the most predictive of sensitivity to ABT-737. Between B cell and Prolif cells, the decrease in Bad sensitivity has a p value of 0.0298 (p<0.05, *) and is therefore significant. Between Prolifs and LCLs, the decrease in Bmf sensitivity has a p value of 0.0796 (p<0.1, ns), which is not statistically significant. However, the biological effect is still meaningful and more apparent when comparing B cells and LCLs, where the decrease in Bmf sensitivity has a p value of 0.0071 (p<0.01, **) and is highly significant.

For overall mitochondrial priming, we applied the same approach to low-dose Puma and Bim treatments. While B cells and Prolifs were not significantly different in their sensitivity to Bim 1µM (p=0.2586, ns), Prolifs and LCLs were significantly different (p=0.0098, **). Similarly, when treated with Puma 10µM, there was no significant difference between B cells and Prolifs (p=0.1532, ns) or between Prolifs and LCLs (p=0.7800, ns). However, there was a significant difference between B cells and LCLs (p=0.0211, *), indeed suggesting that EBV infection decreases overall mitochondrial priming.

These statistical tests will be included in Figure 1 as well as the statistical reporting table.

2) Both BCl-2 and BCl-w have similar characteristics in the BH3 profiles (Figure 1), but why do the authors only focus on the function of BCl-2 in the following experiments? For instance, the average IC_50_ for "Prolif" and "LCL" are completely opposite after treatment with ABT-737 or ABT-199, suggesting unique functions of BCl-xL and BCl-w in these cells (Figure 2). However, only BCl-2 expression was detected in Figure 3. Furthermore, in contrast to WT LCL, Δ3A LCL was sensitive to both Bad and Bmf peptides (Figure 6). Why does this indicate BCl-2 dependence, but not BCl-w dependence?

We agree that with profiling alone it is impossible to tease apart the difference between BCL-2 and BCL-w. This is precisely the reason that we compare the resistance of cells to ABT-737 and ABT-199. To this concern, we disagree that the IC_50_ for ABT-737 and ABT-199 are “completely opposite” in Prolif cells and LCLs. While the average IC_50_ in Prolif cells is higher than LCL treated cells for ABT-199, the IC_50_ in Prolifs to neither ABT-199 nor ABT-737 is significantly different than in LCLs. We will state this explicitly in the figure and text in the resubmission.

With regards to the BH3 Profiling of the Δ3A LCL in Figure 6 being sensitive to both Bad and Bmf peptide treatment, we will address the potential for BCL-w protection by repeating the experiment in Figure 5 with ABT-199.

3) EBV could induce potent resistance to ABT-199, a specific inhibitor of BCL-2, so it would be better to investigate EBV-induced resistance to BCl-2 antagonism with ATB-199, instead of ABT-737. However, of particular note is that the use of the inhibitors can be misleading and cannot be totally substantiated by the results and the conclusion made without further validation.

As mentioned above, we will repeat the experiment in Figure 5 with ABT-199.

4) The expression change of MCL-1 is different from BCl-2/Bfl-1's (Figure 3). Therefore, it seems that MCL-1 and BCl-2/Bfl-1 protect EBV-infected B-cells from apoptosis early after infection and late in infection, respectively. As shown in Figure 1, the survival mechanism may also rely on BCl-2/BCl-w to block Noxa-induced depolarization in LCLs. The authors also noted there was no available BCl-1 antibody (subsection “MCL-1 and BCL-2 protect EBV-infected proliferating B cells from apoptosis early after infection, while BFL-1 additionally protects LCLs from apoptosis late in infection”, first paragraph), actually, Bfl-1 is also known as BCL2A1 (Bcl2 related protein A1), so there should be tons of commercial antibodies at least for Bfl-1 as well as BCl-2. Therefore, Bfl-1 expression should be detected in these studies and is important to support the conclusions. In addition, the Annexin V positivity of ABT-737-treated LCLs in Figure 2 and Figure 3 are inconsistent, so this conclusion is also difficult to accept and is not convincing.

We have tested numerous commercial antibodies directed towards BFL-1(BCL2A1) with no success. Upon the expert suggestion of our collaborators (Dr. Anthony Letai and Dr. Kris Wood) we have decided that gene expression data would have to be sufficient.

With regards to the Annexin V data shown in Figure 2 and Figure 3, we show a basal apoptotic rate at ~5% (no treatment) and ~10% with 1 µM ABT-737, which is consistent in both figures. However, the scales were different because in 2E we were primarily highlighting the sensitivity of B cells to ABT-737.

5) The Caspase 3/7 activity in ABT-737-treated "Prolif" B-cells increases according to the statistical analysis (Figure 2). Why do the authors indicate that they do not in Figure 4 (subsection “Resistance to BCL-2 antagonism is virus specific”)? The Annexin V positivity should increase in "Prolif" B-cells following ABT-737 treatment (Figure 2). It's better to mark t-test results in the figures, but not in separate files.

While the effect of ABT-737 on Prolif, and LCLs as well, was statistically significant, the actual effect size was very small (~5 basal to ~10% ABT-737 treated for AnnV+ and Caspase activity) relative to that of B cells treated prior to proliferation (~5 to ~80%). This is also reflected in the ~200- 400-fold difference in IC_50_ between the early treatment and later treatment or LCL treatment (Figure 2).

Now we state: “Consistently, mitogen-stimulated proliferating B cells had increased Caspase 3/7 activity and Annexin V positivity following ABT-737 treatment (Figure 4) while EBV-infected cells displayed only marginally increased activity above basal levels (Figure 2).”

We will also note the results of the t-test in the figures as well as legends and the statistical table provided with our submission.

6) Figure 5. Are the qPCR results of viral mRNA normalized to B95.8 LCL as the authors indicated? Why are "B-cell" always shown in the figures? And what do the authors mean by "B-cell" in this manuscript? This is not clear. Is it transformed B-cell, proliferation B-cell Burkitt's lymphoma B-cells?

This experiment was performed using purified, uninfected B cells, infected (with either B95.8 or P3HR1) purified B cells, and normalized to the expression level found in B95.8 LCLs set at 100%. The uninfected B cells were included as a negative/background control, and as such it can be excluded from the manuscript if this makes the data easier to interpret. All of these details will be added to the manuscript.

7) In Figure 6, Δ3A LCL expresses more BCl-2 and MCl-1 than WT at the mRNA level, which is different from the statement "similar to WT" (subsection “EBNA3A is required for MCL-1 mitochondrial localization and BFL-1 transcription”, first paragraph) based on your 2-way ANOVA analysis. The loading control is also missing in Figure 6, while the samples are not normalized in Figure 6. Without this information or control data, it's difficult to evaluate this section of the figure.

Instead of the two-way ANOVA analysis that we initially reported, we will provide t-tests on individual mRNA levels between WT and Δ3A LCL. Based on these analyses, we found no significant difference in the levels of BCL-2 and MCL-1 mRNA, but a significant difference for BFL-1 p value of 0.0002 (p<0.001, ***). This will be updated in the figure and the manuscript.

As we are describing the decay of protein following CHX treatment, the control is time 0. We found no difference between the decay rate of MCL-1 in WT and Δ3A LCLs thus indicating that this was not the mechanism by which EBNA3A impacted MCL-1 activity in the cells. If an additional protein half-life is needed, we could add this for GAPDH or Actin as housekeeping controls.

The samples in 6E are normalized to VDAC and WT levels. This is described in the figure legend.

8) If decreased Bfl-1 transcription inΔ3A LCL is related with Bfl-1 TSS (Figure 6), have the authors determined the transcription activity of Bfl-1 promoter? How about the Bfl-1 protein expression inΔ3A LCL?

We did not include a BFL-1 promoter reporter construct because the regulation of transcription is through an enhancer over 40 kb away. Prior work has indicated that LMP1 and EBNA2 are sufficient to induced BFL1 promoter activity in report constructs. However, our new findings indicate that EBNA3A is required for transcriptional regulation in LCLs through a long-distance enhancer, consistent with its activities across the genome as found by large-scale ChIP-seq studies.